# Sample Complexity of Uniform Convergence for Multicalibration

**Eliran Shabat**[*]
Tel Aviv University
shabat.eliran@gmail.com

**Lee Cohen**[*]
Tel Aviv University
leecohencs@gmail.com

**Yishay Mansour**
Tel Aviv University and
Google Research
mansour.yishay@gmail.com

## Abstract

There is a growing interest in societal concerns in machine learning systems, especially in fairness. Multicalibration gives a comprehensive methodology to address group fairness. In this work, we address the multicalibration error and decouple it from the prediction error. The importance of decoupling the fairness metric (multicalibration) and the accuracy (prediction error) is due to the inherent trade-off between the two, and the societal decision regarding the "right tradeoff" (as imposed many times by regulators). Our work gives sample complexity bounds for uniform convergence guarantees of multicalibration error, which implies that regardless of the accuracy, we can guarantee that the empirical and (true) multicalibration errors are close. We emphasize that our results: (1) are more general than previous bounds, as they apply to both agnostic and realizable settings, and do not rely on a specific type of algorithm (such as differentially private), (2) improve over previous multicalibration sample complexity bounds and (3) implies uniform convergence guarantees for the classical calibration error.

## 1 Introduction

Data driven algorithms influence our everyday lives. While they introduce significant achievements in face recognition, to recommender systems and machine translation, they come at a price. When deployed for predicting outcomes that concern individuals, such as repaying a loan, surviving surgery, or skipping bail, predictive systems are prone to accuracy disparities between different social groups that often induce discriminatory results. These significant societal issues arise due to a variety of reasons: problematic analysis, unrepresentative data and even inherited biases against certain social groups due to historical prejudices. At a high level, there are two separate notions of fairness: *individual fairness* and *group fairness*. Individual fairness is aimed to guarantee fair prediction to each given individual, while group fairness aggregates statistics of certain subpopulations, and compares them. There is a variety of fairness notions for group fairness, such as demographic parity, equalized odds, equalized opportunity, and more (see Barocas et al. (2019)). Our main focus would be on multicalibration criteria for group fairness Hebert-Johnson et al. (2018). Multicalibration of a predictor is defined as follows. There is a prespecified set of subpopulations of interest. The predictor returns a value for each individual (which can be interpreted as a probability). The multicalibration requires that for any "large" subpopulation, and for any value which is predicted "frequently" on that subpopulation, the predicted value and average realized values would be close on this subpopulation. Note that calibration addresses the relationship between the predicted and average realized values, and is generally unrelated to the prediction quality. For example, if a population is half positive and half negative, a predictor that predicts for every individual a value of $0.5$ is perfectly calibrated but has poor accuracy. The work of Hebert-Johnson et al. (2018) proposes a specific algorithm to find a multicalibrated predictor and derived its sample complexity. The work of Liu et al. (2018) related the calibration error to the prediction loss, specifically, it bounds the calibration error as a function of the

---

[*]First two authors have equal contribution.

difference between the predictor loss and the Bayes optimal prediction loss. Their bound implies that in a realizable setting, where the Bayes optimal hypothesis is in the class, using ERMyields a vanishing calibration error, but in an agnostic setting this does not hold. With the motivation of fairness in mind, it is important to differentiate between the prediction loss and the calibration error. In many situations, the society (through regulators) might sacrifice prediction loss to improve fairness, and the right trade-off between them may be task dependent. On the other hand, calibration imposes self-consistency, namely, that predicted values and the average realized values should be similar for any protected group. In particular, there is no reason to prefer un-calibrated predictors over calibrated ones, assuming they have the same prediction loss. An important concept in this regard is uniform convergence. We would like to guarantee that the multicalibration error on the sample and the true multicalibration error are similar. This will allow society to rule-out un-calibrated predictors when optimizing over accuracy and other objectives that might depend on the context and the regulator.

Our main results in this work are sample bounds that guarantee uniform convergence of a given class of predictors. We start by deriving a sample bound for the case of a finite hypothesis class, and derive a sample complexity bound which is logarithmic in the size of the hypothesis class. Later, for an infinite hypothesis class, we derive a sample bound that depends on the *graph dimension* of the class (which is an extension of the VC dimension for multiclass predictions). Finally, we derive a lower bound on the sample size required.

Technically, an important challenge in deriving the uniform convergence bounds is that the multicalibration error depends, not only on the correct labeling but also on the predictions by the hypothesis, similar in spirit to the internal regret notion in online learning. We remark that these techniques are suitable to reproduce generalization bounds for other complex measures such as F-score.

We stress that in contrast to previous works that either attained specific efficient algorithms for finding calibrated predictors Hebert-Johnson et al. (2018) or provided tight connections between calibration error and prediction loss (mainly in the realizable case) Liu et al. (2019), we take a different approach. We concentrate on the statistical aspects of generalization bounds rather than algorithmic ones, and similar to much of the generalization literature in machine learning derive generalization bounds over calibration error for *any* predictor class with a finite size or a finite graph dimension.

Nevertheless, our work does have algorithmic implications. For example, similarly to running ERM, running empirical multicalibration risk minimization over a hypothesis class with bounded complexity $\mathcal{H}$ and "large enough" training set, would output a nearly-multicalibrated predictor, assuming one exists. We guarantee that the empirical and true errors of this predictor would be similar, and derive the required sample size either as a function of the logarithm of the size of the predictor class or of its finite graph dimension. Our bounds improve over previous sample complexity bounds and also apply in more general settings (e.g., agnostic learning). So while multicalibration uniform convergence is not formally necessary for learning multicalibrated predictors, the advantage of our approach is that the learner remains with the freedom to choose any optimization objectives or algorithms, and would still get a good estimation of the calibration error. To the best of our knowledge, this also introduces the first uniform convergence results w.r.t. calibration as a general notion (i.e., even not as a fairness notion).

**Related work:** Calibration has been extensively studied in machine learning, statistics and economics Foster & Vohra (1998); Blum & Mansour (2007); Foster & Hart (2018), and as a notion of fairness dates back to the 1960s Cleary (1968). More recently, the machine learning community adapted calibration as an anti-discrimination tool and studied it and the relationship between it and other fairness criteria Chouldechova (2017); Corbett-Davies et al. (2017); Kleinberg et al. (2017); Pleiss et al. (2017); Liu et al. (2017). There is a variety of fairness criteria, other than calibration, which address societal concerns that arise in machine learning. Fairness notions have two major categories. *Individual-fairness*, that are based on similarity metric between individuals and require that similar individuals will be treated similarly Dwork et al. (2012). *Group-fairness*, such as demographic-parity and equalized-odds, are defined with respect to statistics of subpopulations Barocas et al. (2019). Generalization and uniform convergence are well-explored topics in machine learning, and usually assume some sort of hypotheses class complexity measures, such as VC-dimension, Rademacher complexity, Graph-dimension and Natarajan-dimension Ben-David et al. (1995); Daniely et al. (2011); Shalev-Shwartz & Ben-David (2014). In this work we build on these classic measures to derive our bounds. Generalization of fairness criteria is a topic that receives great attention recently. The works of Kim et al. (2018); Yona & Rothblum (2018) define metric notions that are based on Dwork et al. (2012) and derive generalization guarantees. Other works relax the assumption of a known fairness

metric and derive generalization with respect to Individual Fairness based on oracle queries that simulate human judgments Gillen et al. (2018); Bechavod et al. (2020); Ilvento (2020). Bounds for alternative fairness notions, such as equalized-odds, gerrymandering, multi-accuracy, and envy-free appear in Woodworth et al. (2017); Kearns et al. (2018); Kim et al. (2019); Balcan et al. (2019). We remark that this work does not provide generalization bounds for margin classifiers in the context of fairness, and we leave it for future work.

Multicalibration is a group-fairness notion that requires calibration to hold simultaneously on multiple subpopulations Hebert-Johnson et al. (2018). They proposed a polynomial-time differentially-private algorithm that learns a multicalibrated predictor from samples in agnostic setup. A byproduct of their choice of Differently Private algorithm is that their algorithm and analysis is limited to a finite domain. Our work provides generalization uniform convergence bounds that are independent of the algorithm that generates them, and also improve their sample bounds. The work of Liu et al. (2019) bounds the calibration error by the square-root of the gap between its expected loss and the Bayes-optimal loss, for a broad class of loss functions. While in realizable settings this gap is vanishing, in agnostic settings this gap can be substantial. Our results do not depend on the hypothesis' loss to bound the calibration error, which allows us to give guarantees in the agnostic settings as well.

## 2 Model and Preliminaries

Let $\mathcal{X}$ be any finite or countable domain (i.e., $\mathcal{X}$ is a population and each domain point encodes an individual) and let $\{0, 1\}$ be the set of possible *outcomes*. Let $D$ be a probability distribution over $\mathcal{X} \times \{0, 1\}$, i.e., a joint distribution over domain points and their outcomes. Intuitively, given pairs $(x_i, y_i)$, we assume that outcomes $y_i \in \{0, 1\}$ are the realizations of underlying random sampling from independent Bernoulli distributions with (unknown) parameters $p^*(x_i) \in [0, 1]$. The goal of the learner is to predict the (unknown) parameters $p^*(x_i)$, given a domain point $x_i$. Let $\mathcal{Y} \subseteq [0, 1]$ be the set of possible *predictions values*. A *predictor* (hypothesis) $h$ is a function that maps domain points from $\mathcal{X}$ to prediction values $v \in \mathcal{Y}$. A set of predictors $h : \mathcal{X} \to \mathcal{Y}$ is a *predictor class* and denoted by $\mathcal{H}$. Let $\Gamma = \{U_1, ..., U_{|\Gamma|}\}$ be a finite collection of subpopulations (possibly overlapping) from the domain $\mathcal{X}$ (technically, $\Gamma$ is a collection of subsets of $\mathcal{X}$). Throughout this paper, we will distinguish between the case where $\mathcal{Y}$ is a finite subset of $[0, 1]$ and the case where $\mathcal{Y} = [0, 1]$ (continuous). Both cases depart from the classical binary settings where $\mathcal{Y} = \{0, 1\}$, as predictors can return any prediction value $v \in \mathcal{Y}$ (e.g., $v = 0.3$). We define $\Lambda$ to be a *partition* of $\mathcal{Y}$ into a finite number of subsets, that would have different representations in the continuous and finite cases. For the continuous case where $\mathcal{Y} = [0, 1]$, we would partition $\mathcal{Y}$ into a finite set of intervals using a *partition parameter* $\lambda \in (0, 1]$ that would determinate the lengths of the intervals. Namely, $\Lambda_\lambda := \{\{I_j\}_{j=0}^{\frac{1}{\lambda}-1}\}$, where $I_j = [j\lambda, (j+1)\lambda)$. When $\mathcal{Y}$ is finite, $\Lambda$ would be a set of singletons: $\Lambda = \{\{v\} : v \in \mathcal{Y}\}$ and $h(x) \in I = \{v\}$ is equivalent to $h(x) = v$.

**Definition 1** (Calibration error). *The* calibration error *of predictor* $h \in \mathcal{H}$ *w.r.t. a subpopulation* $U \in \Gamma$ *and an interval* $I \subseteq [0, 1]$, *denoted by* $c(h, U, I)$ *is the difference between the expectations of* $y$ *and* $h(x)$, *conditioned on domain points from* $U$ *that* $h$ *maps to values in* $I$. *I.e.,*

$$c(h, U, I) := \mathop{\mathbb{E}}_D [y \mid x \in U, h(x) \in I] - \mathop{\mathbb{E}}_D [h(x) \mid x \in U, h(x) \in I]$$

Notice that for the case where $\mathcal{Y}$ is finite, we can rewrite the expected calibration error as

$$c(h, U, I = \{v\}) = \mathop{\mathbb{E}}_D [y \mid x \in U, h(x) = v] - v$$

Since calibration error of predictors is a measure with respect to a specific pair of subpopulation $U$ and an interval $I$, we would like to have a notion that captures "well-calibrated" predictors on "large enough" subpopulations and "significant enough" intervals $I$ that $h$ maps domain points (individuals) to, as formalized in the following definition.

**Definition 2** (Category). *A* category *is a pair* $(U, I)$ *of a subpopulation* $U \in \Gamma$ *and an interval* $I \in \Lambda$. *We say that a category* $(U, I)$ *is* interesting *according to predictor* $h$ *and parameters* $\gamma, \psi \in (0, 1]$, *if* $\Pr_D[x \in U] \geq \gamma$ *and* $\Pr_D[h(x) \in I \mid x \in U] \geq \psi$.

We focus on predictors with calibration error of at most $\alpha$ for any interesting category.

**Definition 3** $((\alpha, \gamma, \psi)$–multicalibrated predictor). *A predictor* $h \in \mathcal{H}$ *is* $(\alpha, \gamma, \psi)$–multicalibrated, *if for every interesting category* $(U, I)$ *according to* $h$, $\gamma$ *and* $\psi$, *the absolute value of the calibration error of* $h$ *w.r.t. the category* $(U, I)$ *is at most* $\alpha$, *i.e.,* $\left| c(h, U, I) \right| \leq \alpha$.

We define empirical versions for calibration error and $(\alpha, \gamma, \psi)$–multicalibrated predictor.

**Definition 4** (Empirical Calibration error)**.** *Let* $(U, I)$ *be a category and let* $S^m = \{(x_1, y_1), ..., (x_m, y_m)\}$ *be a training set of* $m$ *samples drawn i.i.d. from* $D$*. The* empirical calibration error *of a predictor* $h \in \mathcal{H}$ *w.r.t.* $(U, I)$ *and* $S$ *is:*

$$\hat{c}(h, U, I, S) := \sum_{i=1}^{m} \frac{\mathbb{I}\left[x_i \in U, h(x_i) \in I\right]}{\sum_{j=1}^{m} \mathbb{I}\left[x_j \in U, h(x_j) \in I\right]} y_i - \sum_{i=1}^{m} \frac{\mathbb{I}\left[x_i \in U, h(x_i) \in I\right]}{\sum_{j=1}^{m} \mathbb{I}\left[x_j \in U, h(x_j) \in I\right]} h(x_i),$$

*where* $\mathbb{I}\left[\cdot\right]$ *is the indicator function.*

Notice that when $\mathcal{Y}$ is finite, since $h(x) \in \{v\}$ is equivalent to $h(x) = v$, we can re-write the empirical calibration error as: $\hat{c}(h, U, I = \{v\}, S) := \sum_{i=1}^{m} \frac{\mathbb{I}[x_i \in U, h(x_i) = v]}{\sum_{j=1}^{m} \mathbb{I}[x_j \in U, h(x_j) = v]} y_i - v$.

**Definition 5** $((\alpha, \gamma, \psi)$–Empirically multicalibrated predictor)**.** *A predictor* $h \in \mathcal{H}$ *is* $(\alpha, \gamma, \psi)$*–* empirically multicalibrated *on a sample* $S$ *of i.i.d examples from* $D$*, if for every interesting category* $(U, I)$ *according to* $h$*,* $\gamma$ *and* $\psi$*, we have* $\left|\hat{c}(h, U, I, S)\right| \leq \alpha$.

We assume that the predictors are taken from some predictor class $\mathcal{H}$. Our main goal is to derive sample bounds for the empirical calibration error to "generalize well" for every $h \in \mathcal{H}$ and every interesting category. We formalize it as follows.

**Definition 6** (Multicalibration Uniform Convergence)**.** *A predictor class* $\mathcal{H} \subseteq \mathcal{Y}^{\mathcal{X}}$ *has the* multicalibration uniform convergence *property (w.r.t. collection* $\Gamma$*) if there exist a function* $m_{\mathcal{H}}^{mc}(\epsilon, \delta, \gamma, \psi) \in \mathbb{N}$*, for* $\epsilon, \delta, \gamma, \psi \in (0, 1]$*, such that for every distribution* $D$ *over* $\mathcal{X} \times \{0, 1\}$*, if* $S^m = \{(x_1, y_1), \cdots, (x_m, y_m)\}$ *is a training set of* $m \geq m_{\mathcal{H}}^{mc}(\epsilon, \delta, \gamma, \psi)$ *examples drawn i.i.d. from* $D$*, then for every* $h \in \mathcal{H}$ *and every interesting category* $(U, I)$ *according to* $h$*,* $\gamma$ *and* $\psi$*, the difference between the calibration error and the empirical calibration error is at most* $\epsilon$ *with probability of at least* $1 - \delta$*, i.e.,* $\Pr_D[|\hat{c}(h, U, I, S^m) - c(h, U, I)| \leq \epsilon] > 1 - \delta$.

We emphasize that the property of multicalibration uniform convergence w.r.t. a predictor class $\mathcal{H}$ is neither a necessary nor sufficient for having multicalibrated predictors $h \in \mathcal{H}$. Namely, having uniform convergence property implies only that the empirical and true errors are similar, but does not imply that they are small. In addition, having a predictor with zero multicalibration error (realizability) does not imply anything about the generalization multicalibration error. For example, if $\mathcal{H}$ contains all the possible predictors, there will clearly be a zero empirical error predictor who's true multicalibration error is very high.

When $\mathcal{H}$ is an infinite predictor class, we can achieve generalization by assuming a finite complexity measure. VC-dimension (the definition appears in section A in the Supplementary Material) measures the complexity of binary hypothesis classes. In this work, we rephrase the generalization problem of multicalibration in terms of multiple generalization problems of binary hypothesis classes with finite VC-dimension, and derive sample complexity bounds for it. So our goal is to approximate the (true) calibration error by estimating it on a large sample. Namely, we would like have a property which indicates that a large-enough sample will result a good approximation of the calibration-error for any hypothesis $h \in \mathcal{H}$ and any interesting category $(U, I)$ according to $h$. Our technique for achieving this property uses known results about binary classification. We mention the definitions of "risk function", "empirical-risk function" and "uniform convergence for statistical learning" (the latter appears in section A in the Supplementary Material). For this purpose, $h : \mathcal{X} \to \{0, 1\}$ would denote a binary hypothesis, $\ell : \mathcal{Y} \times \{0, 1\} \to \mathbb{R}_+$, denotes a loss function and $D$ stays a distribution over $\mathcal{X} \times \{0, 1\}$.

**Definition 7** (Risk function, Empirical risk)**.** *The* risk function*, denoted by* $L_D$*, is the expected loss of a hypothesis* $h$ *w.r.t* $D$*, i.e.,* $L_D(h) := \mathbb{E}_{(x,y) \sim D}[\ell(h(x), y)]$*. Given a random sample* $S = ((x_i, y_i))_{i=1}^{m}$ *of* $m$ *examples drawn i.i.d. from* $D$*, the* empirical risk *is the average loss of* $h$ *over the sample* $S$ *i.e.,* $L_S(h) := \frac{1}{m} \sum_{i=1}^{m} \ell(h(x_i), y_i)$.

Note that the definitions of uniform convergence for statistical learning and the multicalibration uniform convergence are distinct. A major difference is that while the notion of uniform convergence for statistical learning imposes a requirement on the risk, which is defined using an expectation over a fixed underlying distribution $D$, the notion of multicalibration uniform convergence imposes a requirement on the calibration error, in which the expectation is over a conditional distribution that depends on the predictor. When the prediction range, $\mathcal{Y}$, is discrete, we consider the standard multiclass complexity notion graph-dimension, which is define as follows.

**Definition 8** (Graph Dimension). *Let $\mathcal{H} \subseteq \mathcal{Y}^{\mathcal{X}}$ be a hypothesis class from domain $\mathcal{X}$ to a finite set $\mathcal{Y}$ and let $S \subseteq \mathcal{X}$. We say that $\mathcal{H}$ G-shatters $S$ if there exists a function $f : S \to \mathcal{Y}$ such that for every $T \subseteq S$ there exists a hypothesis $h \in \mathcal{H}$ such that $\forall x \in S : h(x) = f(x) \iff x \in T$. The graph dimension of $\mathcal{H}$, denoted $d_G(\mathcal{H})$, is the maximal cardinality of a set that is G-shattered by $\mathcal{H}$.*

## 3 Our Contributions

We derive two upper bounds. The first is for a finite predictor class, in which we discretize $\mathcal{Y} = [0, 1]$ into $\Lambda_\lambda$ and derive a bound which depends logarithmicly on $\lambda^{-1}$. We complement our upper bounds with the following lower bound result.

**Theorem 9.** *Let $\mathcal{H} \subseteq \mathcal{Y}^{\mathcal{X}}$ be a finite predictor class. Then, $\mathcal{H}$ has the uniform multicalibration convergence property with $m_{\mathcal{H}}^{mc}(\epsilon, \delta, \gamma, \psi) = O\left(\frac{1}{\epsilon^2 \gamma \psi} \log\left(|\Gamma||\mathcal{H}|/\delta\lambda\right)\right)$.*

**Theorem 10.** *Let $\mathcal{H} \subseteq \mathcal{Y}^{\mathcal{X}}$ be an infinite predictor class from domain $\mathcal{X}$ to a discrete prediction set $\mathcal{Y}$ with finite graph-dimension $d_G(\mathcal{H}) \leq d$, then $\mathcal{H}$ has the uniform multicalibration convergence property with $m_{\mathcal{H}}^{mc}(\epsilon, \delta, \gamma, \psi) = O\left(\frac{1}{\epsilon^2 \psi^2 \gamma}\left(d + \log\left(|\Gamma||\mathcal{Y}|/\delta\right)\right)\right)$.*

**Theorem 11.** *Let $\mathcal{H}$ be a finite predictor class or an infinite predictor class with finite graph-dimension $d_G(\mathcal{H}) \leq d$. Then, $\mathcal{H}$ has multicalibration uniform convergence with $m(\epsilon, \delta, \psi, \gamma) = \Omega(\frac{1}{\psi\gamma\epsilon^2} \ln(1/\delta))$ samples.*

Rewriting the sample bound of Hebert-Johnson et al. (2018) using our parameters, they have $O\left(\frac{1}{\epsilon^3 \cdot \psi^{3/2} \cdot \gamma^{3/2}} \log(\frac{|\Gamma|}{\epsilon \cdot \gamma \cdot \delta})\right)$. Comparing the bounds, the most important difference is the dependency on $\epsilon$, the generalization error. They have a dependency of $\epsilon^{-3}$, while we have of $\epsilon^{-2}$, which is tight due to our lower bound. For the dependency on $\gamma$, they have $\gamma^{-3/2}$, while we have $\gamma^{-1}$, which is also tight. For the dependency on $\psi$, they have $\psi^{-3/2}$, while we have $\psi^{-1}$ for a finite hypothesis class (which is tight due to our lower bound) and $\psi^{-2}$ for an infinite hypothesis class. Finally, recall that the bound of Hebert-Johnson et al. (2018) applies only to their algorithm and since it is a differentially private algorithm, it requires the domain $\mathcal{X}$ to be finite, while our results apply to continuous domains as well. Note that having $(\alpha, \gamma, \psi)-$ empirically multicalibrated predictor on large random sample, guarantees that, with high probability, it is also $(\alpha + \epsilon, \gamma, \psi)$–mutlicalibrated with respect to the underlying distribution, where $\epsilon$ is the generalization error that depends on the sample size. For brevity, we only overview our proof techniques, and provide full proofs in the Supplementary material.

## 4 Finite Predictor Classes

We start by analyzing the case in which $\mathcal{H}$ is finite and the prediction set $\mathcal{Y}$ is continuous. In this setup, we will utilize the fact that a finite $\mathcal{H}$ implies a finite number of categories, i.e., a partition of the population $\mathcal{X}$ into sub-groups according to $\mathcal{H}$, $\Gamma$ and $\Lambda_\lambda$. This fact, using the union-bound, will allow us to translate any confidence we have over a single interesting category, to a confidence over all interesting categories while only suffering a logarithmic increase in the number of possible categories. Recall that in this setup, the prediction-intervals set, $\Lambda$, is a partition of $\mathcal{Y} = [0, 1]$ into a finite set of intervals of length $\lambda$, namely, $\Lambda = \{I_j\}_{j=0}^{\frac{1}{\lambda}-1} = \{[j\lambda, (j+1)\lambda)\}_{j=0}^{\frac{1}{\lambda}-1}$.

Our upper bound analysis will use the following intuition. Assuming a large sample, with high probability, each interesting category would have a "large enough" sub-sample, which would yield a good approximation of it's calibration error with high probability.

**Theorem 9.** *Let $\mathcal{H} \subseteq \mathcal{Y}^{\mathcal{X}}$ be a finite predictor class. Then, $\mathcal{H}$ has the uniform multicalibration convergence property with $m_{\mathcal{H}}^{mc}(\epsilon, \delta, \gamma, \psi) = O\left(\frac{1}{\epsilon^2 \gamma \psi} \log\left(|\Gamma||\mathcal{H}|/\delta\lambda\right)\right)$.*

In the proof of Theorem 9 we use the relative Chernoff inequality (Lemma 23) and union bound to guarantee,with probability of at least $1 - \delta/2$, a large sub-sample for every predictor $h \in \mathcal{H}$ and for every interesting category $(U, I)$ according to $h$. Then, we use the absolute Chernoff inequality (Lemma 22) to show that with probability at least $1 - \delta/2$, for every $h \in \mathcal{H}$ and every interesting category $(U, I)$ according to $h$, the empirical calibration error does not deviate from the (true) calibration error by more than $\epsilon$. The following corollary indicates that having $(\alpha, \gamma, \psi)-$ empirically multicalibrated predictor on a large random sample, guarantees it is also $(\alpha + \epsilon, \gamma, \psi)$–mutlicalibrated

with respect to the underlying distribution with high probability, where $\epsilon$ is a generalization error that depends on the sample size. It follows immediately from Theorem 9.

**Corollary 12.** *Let $\mathcal{H} \subseteq \mathcal{Y}^{\mathcal{X}}$ be a finite predictor class and let $D$ be a distribution over $\mathcal{X} \times \{0,1\}$. Let $S$ be a random sample of $m$ examples drawn i.i.d. from $D$ and let $h \in \mathcal{H}$ be $(\alpha, \gamma, \psi)$– empirically multicalibrated predictor on $S$. Then, for any $\epsilon, \delta \in (0,1)$, if $m \geq \frac{8}{\epsilon^2 \gamma \psi} \log \left( \frac{8|\Gamma||\mathcal{H}|}{\delta \lambda} \right)$, then with probability at least $1 - \delta$, $h$ is $(\alpha + \epsilon, \gamma, \psi)$–multicalibrated w.r.t. the underlying distribution $D$.*

## 5   Predictor Classes with Finite Graph Dimension

Throughout this section we assume that the predictions set $\mathcal{Y}$ is discrete. This assumption allows us to analyze the multicalibration generalization of possibly infinite hypothesis classes with finite known multiclass complexity measures such as the graph-dimension. (We discuss the case of $\mathcal{Y} = [0,1]$ at the end of the section.) Recall that in this setup, the prediction-intervals set, $\Lambda$, contains singleton intervals with values taken from $\mathcal{Y}$, namely, $\Lambda = \{\{v\} \mid v \in \mathcal{Y}\}$. Thus, if a prediction, $h(x)$ is in the interval $\{v\}$, it means the prediction value is exactly $v$, i.e., $h(x) \in \{v\} \Leftrightarrow h(x) = v$. As we have mentioned earlier, part of our technique is to reduce multicalibration generalization to the generalization analysis of multiple binary hypothesis classes to get sample complexity bounds. The Fundamental Theorem of Statistical Learning (see Theorem 26, Section A in the Supplementary material) provides tight sample complexity bounds for uniform convergence for binary hypothesis classes. A direct corollary of this theorem indicates that by using "large enough" sample, the difference between the true probability to receive a positive outcome and the estimated proportion of positive outcomes, is small, with high probability.

**Corollary 13.** *Let $\mathcal{H} \subseteq \{0,1\}^{\mathcal{X}}$ be a binary hypothesis class with $VCdim(\mathcal{H}) \leq d$. Then, there exists a constant $C \in \mathbb{R}$ such that for any distribution $D$, and parameters $\epsilon, \delta \in (0,1)$, if $S = \{x_i, y_i\}_{i=1}^m$ is a sample of $m$ i.i.d. examples from $D$, and $m \geq C((d + \log(1/\delta))/\epsilon^2)$ then with probability at least $1 - \delta$, $\forall h \in \mathcal{H} : \left| \frac{1}{m} \sum_{i=1}^m h(x_i) - \Pr_{x \sim D}[h(x) = 1] \right| < \epsilon$.*

Before we move on, we want to emphasize the main technical challenge in deriving generalization bounds for infinite predictor classes. Unlike PAC learning, in multicalibration learning the distribution over the domain is dependent on the predictors class. Each pair of $h \in \mathcal{H}, v \in \mathcal{Y}$ induce a distribution over the domain points $x$ such that $h(x) = v$. As the number of predictors in the class is infinite, we cannot apply a simple union bound over the various induced distributions. This is a main challenge in our proof. In order to utilize the existing theory about binary hypothesis classes we have to represent the calibration error in terms of binary predictors. For this purpose, we define the notion of "binary predictor class", $\mathcal{H}_v \subseteq \{0,1\}^{\mathcal{X}}$, that depends on the original predictor class $\mathcal{H}$ and on a given prediction value $v \in \mathcal{Y}$. Each binary predictor $h_v \in \mathcal{H}_v$ corresponds to a predictor $h \in \mathcal{H}$ and value $v \in \mathcal{Y}$ and predicts 1 on domain points $x$ if $h$ predicts $v$ on them (and 0 otherwise).

**Definition 14** (Binary Predictor). *Let $h \in \mathcal{H}$ be a predictor and let $v \in \mathcal{Y}$ be a prediction value. The binary predictor of $h$ and $v$, denoted $h_v(x)$, is the binary function that receives $x \in \mathcal{X}$ and outputs 1 iff $h(x) = v$, i.e., $h_v(x) = \mathbb{I}[h(x) = v]$. The binary predictor class w.r.t. the original predictor class $\mathcal{H}$ and value $v \in \mathcal{Y}$, denoted by $\mathcal{H}_v$, is defined as $\mathcal{H}_v = \{h_v : h \in \mathcal{H}\}$.*

The definition of binary predictors alone is not sufficient since it ignores the outcomes $y \in \{0,1\}$. Thus, we define true positive function, $\phi_{h_v} \in \Phi_{\mathcal{H}_v}$, that corresponds to a binary predictor $h_v$, such that given a pair $(x \in \mathcal{X}, y \in \{0,1\})$, it outputs 1 iff $h_v(x) = 1$ and $y = 1$.

**Definition 15** (True positive function). *Let $\mathcal{H}_v \subseteq \{0,1\}^{\mathcal{X}}$ be a binary predictor class and let $h_v \in \mathcal{H}_v$ be a binary predictor. Then, the true positive function w.r.t. $h_v$ is $\phi_{h_v}(x,y) := \mathbb{I}[h_v(x) = 1, y = 1]$. The true positive class of $\mathcal{H}_v$, is defined $\Phi_{\mathcal{H}_v} := \{\phi_{h_v} : h_v \in \mathcal{H}_v\}$.*

Using the above definitions we can re-write the calibration error as follows. Let $I_v = \{v\}$ be a singleton interval. Then, the calibration error and the empirical calibration errors take the following forms:
$c(h, U, I_v) = \mathbb{E}_D[y \mid x \in U, h(x) = v] - v = \Pr_{(x,y) \sim D}[y = 1 \mid x \in U, h(x) = v] - v$.

$$\hat{c}(h, U, I_v, S) = \sum_{i=1}^m \frac{\mathbb{I}[x_i \in U, h(x_i) = v]}{\sum_{j=1}^m \mathbb{I}[x_j \in U, h(x_j) = v]} y_i - v = \frac{\sum_{i=1}^m \mathbb{I}[x_i \in U, h(x_i) = v, y_i = 1]}{\sum_{j=1}^m \mathbb{I}[x_j \in U, h(x_j) = v]} - v.$$

The probability term in the calibration error notion is conditional on the subpopulation $U \in \Gamma$ and on the prediction value $h(x)$. Thus, different subpopulations and different predictors induce different

distributions on the domain $\mathcal{X}$. To understand the challenge, consider the collection of conditional distributions induced by $h \in \mathcal{H}$ and an interesting category $(U, I)$. Since $\mathcal{H}$ is infinite, we have an infinite collection of distributions, and guaranteeing uniform convergence for such a family of distributions is challenging. In order to use the fundamental theorem of learning (Theorem 26), we circumvent this difficulty by re-writing the calibration error as follows.

$$c(h, U, I_v) = \Pr_{(x,y) \sim D} [y = 1 \mid x \in U, h(x) = v] - v = \frac{\Pr[y = 1, h(x) = v \mid x \in U]}{\Pr[h(x) = v \mid x \in U]} - v.$$

Later , we will separately approximate the numerator and denominator.

Finally, we use the definitions of binary predictor, $h_v$, and true positive functions $\phi_{h_v}$, to represent the calibration error in terms of binary functions. Thus, the calibration error and the empirical calibration error take the following forms:

$$c(h, U, I_v) = \frac{\Pr[y = 1, h(x) = v \mid x \in U]}{\Pr[h(x) = v \mid x \in U]} - v = \frac{\Pr[\phi_{h_v}(x, y) = 1 \mid x \in U]}{\Pr[h_v(x) = 1 \mid x \in U]} - v,$$

$$\hat{c}(h, U, I_v, S) = \frac{\sum_{i=1}^m \mathbb{I}[x_i \in U, h(x_i) = v, y_i = 1]}{\sum_{j=1}^m \mathbb{I}[x_j \in U, h(x_j) = v]} - v = \frac{\sum_{i=1}^m \mathbb{I}[x_i \in U, \phi_{h_v}(x_i, y_i) = 1]}{\sum_{j=1}^m \mathbb{I}[x_j \in U, h_v(x_j) = 1]} - v.$$

Since the calibration error as written above depends on binary predictors, if we can prove that the complexity of the hypothesis classes containing them has finite VC-dimension, then we will be able to approximate for each term separately. Recall that in this section we are dealing with multiclass predictors, which means that we must use multiclass complexity notion. We analyze the generalization of calibration by assuming that the predictor class $\mathcal{H}$ has a finite graph-dimension. The following lemma states that a finite graph dimension of $\mathcal{H}$ implies finite VC-dimension of the binary prediction classes $\mathcal{H}_v$ for any $v \in \mathcal{Y}$. This result guarantees good approximation for the denominator term, $\Pr[h_v(x) = 1 \mid x \in U]$, in the calibration error. We remark that while the following lemma is also a direct corollary when considering graph dimension as a special case of Psi-dimension Ben-David et al. (1995), for completeness, we provide a simple proof for in the Supplementary material.

**Lemma 16.** *Let $\mathcal{H} \subseteq \mathcal{Y}^{\mathcal{X}}$ be a predictor class such that $d_G(\mathcal{H}) \leq d$. Then, for any $v \in \mathcal{Y}$, $VCdim(\mathcal{H}_v) \leq d$.*

In addition to the complexity bound of the binary predictor classes $\mathcal{H}_v$, we would like to derive a bound on the VC-dimension of the prediction-outcome classes $\Phi_{\mathcal{H}_v}$ which would enable a good approximation of the numerator term, $\Pr[\phi_{h_v}(x, y) = 1 \mid x \in U]$ in the calibration error. This bound is achieved by using the following lemma that indicates that the VC-dimension of $\Phi_{\mathcal{H}_v}$ is bounded by the VC-dimension of $\mathcal{H}_v$.

**Lemma 17.** *Let $\mathcal{H}_v \subseteq \{0, 1\}^{\mathcal{X}}$ be a binary predictor class with $VCdim(\mathcal{H}_v) \leq d$, and let $\Phi_{\mathcal{H}_v}$ be the true positive class w.r.t. $\mathcal{H}_v$. Then, $VCdim(\Phi_{\mathcal{H}_v}) \leq d$.*

The fact that the VC-dimensions of $\mathcal{H}_v$ and $\Phi_{\mathcal{H}_v}$ are bounded enables to utilize the existing theory and derive sampling bounds for accurate approximations for the numerator and the denominator of the calibration error with high probability, respectively. Lemma 18 formalizes these ideas.

**Lemma 18.** *Let $\mathcal{H} \subseteq \mathcal{Y}^{\mathcal{X}}$ be a predictor class with $d_G(\mathcal{H}) \leq d$. Let $v \in \mathcal{Y}$ be a prediction value and let $U \subset \mathcal{X}$ be a subpopulation. Then, there exist a constant $C \in \mathbb{R}$ such that for any distribution $D$ over $\mathcal{X} \times \{0, 1\}$ and $\epsilon, \delta \in (0, 1)$, if $D_U$ is the induced distribution on $U \times \{0, 1\}$ and $S = \{x_i, y_i\}_{i=1}^m$ is a random sample of size $m \geq C \frac{d + \log(1/\delta)}{\epsilon^2}$ drawn i.i.d. according to $D_U$, then with probability at least $1 - \delta$ for every $h \in \mathcal{H}$:*

$$\left| \frac{1}{m} \sum_{i=1}^m \mathbb{I}[h(x_i) = v] - \Pr_{D_U}[h(x) = v] \right|, \left| \frac{1}{m} \sum_{i=1}^m \mathbb{I}[h(x_i) = v, y = 1] - \Pr_{D_U}[h(x) = 1, y = 1] \right| \leq \epsilon.$$

Having an accurate approximation of the denominator and numerator terms of the calibration error does not automatically implies good approximation for it. For example, any approximation error in the numerator is scaled by $1$ divided by the denominator's value. The following lemma tells us how accurate the approximations of the numerator and the denominatorshould be in order to achieve good approximation of the entire fraction, given a lower bound on the true value of the denominator.

**Lemma 19.** *Let $p_1, p_2, \tilde{p}_1, \tilde{p}_2, \epsilon, \psi \in [0, 1]$ such that $p_1, \psi \leq p_2$ and $|p_1 - \tilde{p}_1|, |p_2 - \tilde{p}_2| \leq \psi \epsilon / 3$. Then, $|p_1/p_2 - \tilde{p}_1/\tilde{p}_2| \leq \epsilon$.*

Since multicalibration uniform convergence requires empirical calibration errors of interesting categories to be close to their respective (true) calibration errors, a necessary condition is to have a large sample from every large subpopulation $U \in \Gamma$. The following lemma indicates the sufficient sample size to achieve a large subsample from every large subpopulation with high probability.

**Lemma 20.** *Let $\gamma \in (0,1)$ and let $\Gamma_\gamma = \{U \in \Gamma \mid \Pr_{x \sim D}[x \in U] \geq \gamma\}$ be the collection of subpopulations from $\Gamma$ that has probability at least $\gamma$ according to $D$. Let $\delta \in (0,1)$ and let $S = \{(x_i, y_i)\}_{i=1}^m$ be a random sample of $m$ i.i.d. examples from $D$. Then, with probability at least $1 - \delta$, if $m \geq \frac{8}{\gamma} \log(|\Gamma|/\delta)$, it holds that $\forall U \in \Gamma_\gamma : |S \cap U| > \frac{\gamma m}{2}$.*

The following theorem combines all the intuition described above and prove an upper bound on the sample size needed to achieve multicalibration uniform convergence. It assumes that the predictor class $\mathcal{H}$ has a finite graph-dimension, $d_G(\mathcal{H})$ and uses Lemma 16 and Lemma 17 to derive an upper bound on the VC-dimension of $\mathcal{H}_v$ and $\Phi_{\mathcal{H}_v}$. Then, it uses Lemma 18 to bound the sample complexity for "good" approximation of the numerator and the denominator of the calibration error.

**Theorem 10.** *Let $\mathcal{H} \subseteq \mathcal{Y}^{\mathcal{X}}$ be an infinite predictor class from domain $\mathcal{X}$ to a discrete prediction set $\mathcal{Y}$ with finite graph-dimension $d_G(\mathcal{H}) \leq d$, then $\mathcal{H}$ has the uniform multicalibration convergence property with $m_{\mathcal{H}}^{mc}(\epsilon, \delta, \gamma, \psi) = O\left(\frac{1}{\epsilon^2 \psi^2 \gamma}\left(d + \log(|\Gamma||\mathcal{Y}|/\delta)\right)\right)$.*

The proof of Theorem 10 uses the relative Chernoff bound (Lemma 23) to show that with probability at least $1 - \delta/2$, every subpopulation $U \in \Gamma$ with $\Pr_D[U] \geq \gamma$, has a sub-sample of size at least $\frac{\gamma m}{2}$, namely $|S \cap U| \geq \frac{\gamma m}{2}$. Then, it uses Lemmas 16 and 17 to show that for every $v \in \mathcal{Y}$, $VCdim(\Phi_{\mathcal{H}_v}) \leq VCdim(\mathcal{H}_v) \leq d_G(\mathcal{H})$. It proceeds by applying Lemma 18 to show that, with probability at least $1 - \delta/2$, for every prediction value $v \in \mathcal{Y}$ and every subpopulation $U \in \Gamma$, if

$$|S \cap U| \geq \frac{\gamma m}{2}, \text{ then: } \left|\Pr\left[\phi_{h_v}(x,y) = 1 \mid x \in U\right] - \frac{1}{|S \cap U|}\sum_{i=1}^m \mathbb{I}\left[x_i \in U, \phi_{h_v}(x_i, y_i) = 1\right]\right| \leq \frac{\psi \epsilon}{3},$$

and $\left|\Pr\left[h_v(x) = 1 \mid x \in U\right] - \frac{1}{|S \cap U|}\sum_{j=1}^m \mathbb{I}\left[x_j \in U, h_v(x_j) = 1\right]\right| \leq \frac{\psi \epsilon}{3}$.

Finally, it concludes the proof of Theorem 10 using Lemma 19.   Similarly to the discussion of Corollary 12, we derive the following corollary from Theorem 10.

**Corollary 21.** *Let $\mathcal{H} \subseteq \mathcal{Y}^{\mathcal{X}}$ be a predictor class with $d_G(\mathcal{H}) \leq d$ and let $D$ be a distribution over $\mathcal{X} \times \{0,1\}$. Let $S$ be a random sample of $m$ examples drawn i.i.d. from $D$ and let $h \in \mathcal{H}$ be $(\alpha, \gamma, \psi)-$ empirically multicalibrated predictor on $S$. Then, there exists a constant $C > 0$ such that for any $\epsilon, \delta \in (0,1)$, if $m \geq \frac{C}{\epsilon^2 \psi^2 \gamma}(d + \log(|\Gamma||\mathcal{Y}|/\delta))$, then with probability at least $1 - \delta$, $h$ is $(\alpha + \epsilon, \gamma, \psi)-$multicalibrated w.r.t. the underlying distribution $D$.*

**Finite versus continuous $\mathcal{Y}$:** We have presented all the results for the infinite predictor class using a finite prediction-interval set $\Lambda = \{\{v\}|v \in \mathcal{Y}\}$. We can extend our results to the continuous $\mathcal{Y} = [0,1]$ in a straightforward way. We can simply round the predictions to a value $j\lambda$, and there are $1/\lambda$ such values. This will result in an increase in the calibration error of at most $\lambda$. (Note that in the finite predictor class case, we have a more refine analysis that does not increase the calibration error by $\lambda$.) The main issue with this approach is that the graph-dimension depends on the parameter $\lambda$ through the induced values $j\lambda$. Since we select $\lambda$ and the points $j\lambda$, the magnitude of graph-dimension depends not only on the predictor class but also on parameters which are in our control, and therefore harder to interpret. For this reason we preferred to present our results for the finite $\mathcal{Y}$ case, and remark that one can extend them to the continuous $\mathcal{Y} = [0,1]$ case.

# 6   Lower Bound

We prove a lower bound for the required number of samples to get multicalibration uniform convergence. The proof is done by considering a predictor class with a single predictor that maps $\gamma\psi$ fraction of the population to $1/2 + \epsilon$. We show that this class has multicalibration uniform convergence property for $1/2 + \epsilon$ and then show how to use this property to distinguish between biased coins, which yield a lower bound of $\Omega(\frac{1}{\psi\gamma\epsilon^2}\ln(1/\delta))$ on the sample complexity.

**Theorem 11.** *Let $\mathcal{H}$ be a finite predictor class or an infinite predictor class with finite graph-dimension $d_G(\mathcal{H}) \leq d$. Then, $\mathcal{H}$ has multicalibration uniform convergence with $m(\epsilon, \delta, \psi, \gamma) = \Omega(\frac{1}{\psi\gamma\epsilon^2}\ln(1/\delta))$ samples.*

## Broader Impact

The generalization bounds we derive in this work can be utilized to measure the calibration error of predictors. When it comes to predictors that assign probabilities to individuals (e.g., probability of repaying a loan), verifying that predictors are "multicalibrated" provides society with fairness guarantees about their performances.

While our work makes an important contribution to this significant societal matter, we point out some limitations. First, it is up to regulators to decide upon the subpopulations collection properly. For example, if subpopulations that are characterized by gender are excluded, it might harm major parts of the population. Second, even if the aforementioned issue is tackled, and good approximations for multicalibration are obtained, they do not dismiss other forms of fairness, and as a result, fairness as a general concept is still not guaranteed. In particular, the tools that derive from our work are not applicable for "negligible size" subpopulations (that is, $U \in \Gamma$ such that $\Pr_D[x \in U] < \gamma$). At the extreme, we might have subpopulations which are singletons (single individuals), for which one should use individual fairness rather than group fairness. In addition, regulators should use the tools we present with caution as unforeseeable trade-offs between other fairness constraints might also arise, and insuring one form of fairness could lead to worsening others.

Overall, we emphasize that regulators should heavily consider the full meaning of their choices, and understanding them with respect to multicalibration is only a part of it.

## Acknowledgments

This project has received funding from the European Research Council (ERC) under the European Union's Horizon 2020 research and innovation program (grant agreement No. 882396), and by the Israel Science Foundation (grant number 993/17). Lee Cohen is a fellow of the Ariane de Rothschild Women Doctoral Program.

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
