[Supplementary Material]

# A    Useful Definitions & Theorems

Throughout this paper, we use the following standard Chernoff bounds.

**Lemma 22** (Absolute Chernoff Bound). *Let $X_1, ..., X_n$ be i.i.d. binary random variables with $\mathbb{E}[X_i] = \mu$ for all $i \in [n]$. Then, for any $\epsilon > 0$: $\Pr\left[\left|\frac{1}{n}\sum_{i=1}^n X_i - \mu\right| \geq \epsilon\right] \leq 2\exp(-2\epsilon^2 n)$.*

**Lemma 23** (Relative Chernoff Bound). *Let $X_1, ..., X_n$ be i.i.d. binary random variables and let $X$ denote their sum. Then, for any $\epsilon \in (0,1)$: $\Pr\left[X \leq (1-\epsilon)\mathbb{E}[X]\right] \leq \exp(-\epsilon^2 \mathbb{E}[X]/2)$.*

Next, the definition of Vapnik–Chervonenkis dimension, following by Uniform convergence for statistical learning and the Fundamental Theorem of Statistical Learning.

**Definition 24.** *[VC-dimension] Let $\mathcal{H} \subseteq \{0,1\}^{\mathcal{X}}$ be a hypothesis class. A subset $S = \{x_1, ..., x_{|S|}\} \subseteq \mathcal{X}$ is shattered by $\mathcal{H}$ if: $\left|\left\{\left(h(x_1), ..., h(x_{|S|})\right) : h \in \mathcal{H}\right\}\right| = 2^{|S|}$. The VC-dimension of $\mathcal{H}$, denoted $VCdim(\mathcal{H})$, is the maximal cardinality of a subset $S \subseteq \mathcal{X}$ shattered by $\mathcal{H}$.*

**Definition 25** (Uniform convergence for statistical learning). *Let $\mathcal{H} \subseteq \mathcal{Y}^{\mathcal{X}}$ be a hypothesis class. We say that $\mathcal{H}$ has the uniform convergence property w.r.t. loss function $\ell$ if there exists a function $m_{\mathcal{H}}^{sl}(\epsilon, \delta) \in \mathbb{N}$ such that for every $\epsilon, \delta \in (0,1)$ and for every probability distribution $D$ over $\mathcal{X} \times \{0,1\}$, if $S$ is a sample of $m \geq m_{\mathcal{H}}^{sl}(\epsilon, \delta)$ examples drawn i.i.d. from to $D$, then, with probability of at least $1 - \delta$, for every $h \in \mathcal{H}$, the difference between the risk and the empirical risk is at most $\epsilon$. Namely, with probability $1 - \delta$, $\forall h \in \mathcal{H} : |L_S(h) - L_D(h)| \leq \epsilon$.*

**Theorem 26.** *[The Fundamental Theorem of Statistical Learning] Let $\mathcal{H} \subseteq \{0,1\}^{\mathcal{X}}$ be a binary hypothesis class with $VCdim(\mathcal{H}) = d$ and let the loss function, $\ell$, be the $0-1$ loss. Then, $\mathcal{H}$ has the uniform convergence property with sample complexity $m_{\mathcal{H}}^{UC}(\epsilon, \delta) = \Theta\left(\frac{1}{\epsilon^2}\left(d + \log(1/\delta)\right)\right)$.*

# B    Proofs for Section 4

*Proof.* (Proof of Theorem 9)
Let $S^m = \{(x_1, y_1), ..., (x_m, y_m)\}$ be a random sample of size $m \geq m_{\mathcal{H}}(\epsilon, \delta, \psi, \gamma, \lambda)$ labeled examples drawn i.i.d. according to $D$.

For convenience, throughout the proof we use the following notations. We first define the quantities with respect to the distribution. For a given hypothesis $h \in H$, group $U \in \Gamma$ and interval $I \in \Lambda$, we are interested in the subpoppulation which belongs to $U$ and for which $h$ prediction is in $I$, i.e., $[x \in U, h(x) \in I]$. For this subpoppulation we define: $p(h, U, I)$ the probability of being in this subpoppulation, $\mu_y(h, U, I)$ the average $y$ value in the subpoppulation, and $\mu_h(h, U, I)$, the average prediction, i.e., $h(x)$. The three measures are with respect to the true distribution $D$. Formally,

$$p(h, U, I) := \Pr_D[x \in U, h(x) \in I]$$

$$\mu_y(h, U, I) := \mathbb{E}_D[y \mid x \in U, h(x) \in I]$$

$$\mu_h(h, U, I) := \mathbb{E}_D[h(x) \mid x \in U, h(x) \in I]$$

Similarly we denote the three empirical quantities with respect to the sample. Namely, we denote by $\hat{n}(h, U, I, S)$, $\hat{\mu}_y(h, U, I, S)$ and $\hat{\mu}_h(h, U, I, S)$ the number of samples, empirical outcome and empirical prediction, of the subpoppulation $[x \in U, h(x) \in I]$. Formally,

$$\hat{n}(h, U, I, S) := \sum_{i=1}^m \mathbb{I}[x_i \in U, h(x_i) \in I]$$

$$\hat{\mu}_y(h, U, I, S) := \sum_{i=1}^m \frac{\mathbb{I}[x_i \in U, h(x_i) \in I]}{\hat{n}(h, I, U, S)} y_i$$

$$\hat{\mu}_h(h, U, I, S) := \sum_{i=1}^m \frac{\mathbb{I}[x_i \in U, h(x_i) \in I]}{\hat{n}(h, I, U, S)} h(x_i)$$

Then, the calibration error and the empirical calibration error can be expressed as:

$$c(h, U, I) = \mu_y(h, U, I) - \mu_h(h, U, I)$$
$$\hat{c}(h, U, I, S) = \hat{\mu}_y(h, U, I, S) - \hat{\mu}_h(h, U, I, S)$$

Let $C_h$ denote the collection of all interesting categories according to predictor $h$, namely,

$$C_h := \left\{ (U, I) : U \in \Gamma, I \in \Lambda, \Pr_D[x \in U] \geq \gamma, \Pr_D\left[h(x) \in I \mid x \in U\right] \geq \psi \right\}$$

Note that every interesting category $(U, I) \in C_h$ has a probability of at least $\gamma\psi$, namely, for every $h \in \mathcal{H}$ and for any interesting category $(U, I) \in C_h$:

$$\Pr_{x \sim D}[x \in U, h(x) \in I] = \Pr_{x \sim D}\left[h(x) \in I \mid x \in U\right] \cdot \Pr_{x \sim D}[x \in U] \geq \gamma\psi$$

We define a "bad" event $B^m$ over the samples, as the event there exist some predictor and some interesting category for which the generalization error is larger than $\epsilon$.

$$B^m := \left\{ S \in (\mathcal{X} \times \{0,1\})^m : \exists h \in \mathcal{H}, \exists (U, I) \in C_h : |\hat{c}(h, U, I, S) - c(h, U, I)| > \epsilon \right\}$$

Bounding the probability that $S^m \in B^m$ by $\delta$ implies the theorem. In order to do so, we would like to have a "large enough" induced sample in every interesting category. For this purpose, we define the "good" event, $G^{m,l}$, as the event that indicates that for every predictor, each interesting category has at least $l$ samples.

$$G^{m,l} := \left\{ S \in (\mathcal{X} \times \{0,1\})^m : \forall h \in \mathcal{H}, \forall (U, I) \in C_h : \hat{n}(h, U, I, S) \geq l \right\}$$

We will later set $l$ to achieve $\epsilon$-accurate approximation with confidence $\delta$ later. Note that $G^{m,l}$ is not the complement of $B^m$.

According to the law of total probability the following holds:

$$\Pr[B^m] = \Pr\left[B^m \mid G^{m,l}\right] \Pr\left[G^{m,l}\right] + \Pr\left[B^m \mid \overline{G^{m,l}}\right] \Pr\left[\overline{G^{m,l}}\right]$$

$$\leq \Pr\left[B^m \mid G^{m,l}\right] + \Pr\left[\overline{G^{m,l}}\right]$$

We would like to bound each of the probabilities $\Pr\left[B^m \mid G^{m,l}\right]$ and $\Pr[\overline{G^{m,l}}]$ by $\delta/2$, in order to bound the probability of $B^m$ by $\delta$. We start by bounding $\Pr\left[S^m \in B^m \mid S^m \in G^{m,l}\right]$. By using the union bound:

$$\Pr\left[S^m \in B^m \mid S^m \in G^{m,l}\right]$$

$$= \Pr\left[\exists h \in \mathcal{H}, \exists (U, I) \in C_h : |\hat{c}(h, U, I, S^m) - c(h, U, I)| > \epsilon \;\middle|\; \forall h \in \mathcal{H}, \forall (U, I) \in C_h : \hat{n}(h, U, I, S^m) \geq l\right]$$

$$\leq \sum_{h \in \mathcal{H}} \sum_{(U,I) \in C_h} \Pr\left[|\hat{c}(h, U, I, S^m) - c(h, U, I)| > \epsilon \;\middle|\; \forall h \in \mathcal{H}, \forall (U, I) \in C_h : \hat{n}(h, U, I, S^m) \geq l\right]$$

$$= \sum_{h \in \mathcal{H}} \sum_{(U,I) \in C_h} \Pr\left[|\hat{c}(h, U, I, S^m) - c(h, U, I)| > \epsilon \;\middle|\; \hat{n}(h, U, I, S^m) \geq l\right]$$

By using the triangle inequality:

$$\sum_{h \in \mathcal{H}} \sum_{(U,I) \in C_h} \Pr\left[|\hat{c}(h, U, I, S^m) - c(h, U, I)| > \epsilon \;\middle|\; \hat{n}(h, U, I, S^m) \geq l\right]$$

$$= \sum_{h \in \mathcal{H}} \sum_{(U,I) \in C_h} \Pr\left[|\hat{\mu}_y(h, U, I, S^m) - \hat{\mu}_h(h, U, I, S^m) - \mu_y(h, U, I) + \mu_h(h, U, I)| > \epsilon \;\middle|\; \hat{n}(h, U, I, S^m) \geq l\right]$$

$$\leq \sum_{h \in \mathcal{H}} \sum_{(U,I) \in C_h} \Pr\left[|\hat{\mu}_h(h, U, I, S^m) - \mu_h(h, U, I)| + |\mu_y(h, U, I) - \hat{\mu}_y(h, U, I, S^m)| > \epsilon \;\middle|\; \hat{n}(h, U, I, S^m) \geq l\right]$$

Since $a + b \geq \epsilon$ implies that either $a \geq \epsilon/2$ or $b \geq \epsilon/2$:

$$\sum_{h \in \mathcal{H}} \sum_{(U,I) \in C_h} \Pr\left[ |\hat{\mu}_h(h,U,I,S^m) - \mu_h(h,U,I)| + |\mu_y(h,U,I) - \hat{\mu}_y(h,U,I,S^m)| > \epsilon \;\middle|\; \hat{n}(h,U,I,S^m) \geq l \right]$$

$$\leq \sum_{h \in \mathcal{H}} \sum_{(U,I) \in C_h} \Pr\left[ |\hat{\mu}_h(h,U,I,S^m) - \mu_h(h,U,I)| > \frac{\epsilon}{2} \;\vee\; |\mu_y(h,U,I) - \hat{\mu}_y(h,U,I,S^m)| > \frac{\epsilon}{2} \;\middle|\; \hat{n}(h,U,I,S^m) \geq l \right]$$

And by using the union-bound once again:

$$\sum_{h \in \mathcal{H}} \sum_{(U,I) \in C_h} \Pr\left[ |\hat{\mu}_h(h,U,I,S^m) - \mu_h(h,U,I)| > \frac{\epsilon}{2} \;\vee\; |\mu_y(h,U,I) - \hat{\mu}_y(h,U,I,S^m)| > \frac{\epsilon}{2} \;\middle|\; \hat{n}(h,U,I,S^m) \geq l \right]$$

$$\leq \sum_{h \in \mathcal{H}} \sum_{(U,I) \in C_h} \Pr\left[ |\hat{\mu}_h(h,U,I,S^m) - \mu_h(h,U,I)| > \frac{\epsilon}{2} \;\middle|\; \hat{n}(h,U,I,S^m) \geq l \right]$$

$$+ \Pr\left[ |\mu_y(h,U,I) - \hat{\mu}_y(h,U,I,S^m)| > \frac{\epsilon}{2} \;\middle|\; \hat{n}(h,U,I,S^m) \geq l \right]$$

We would like to use Chernoff inequality (Lemma 22) to bound the probability with a confidence of $1 - \delta/2$. However, in order to do so, we must fix the number of samples, $\hat{n}(h,U,I,S^m)$, that $h$ maps to a certain category (rather than using a random variable). Note that for $\hat{n}(h,U,I,S^m) \geq l$ the probability is maximized at $\hat{n}(h,U,I,S^m) = l$, so we will assume that $\hat{n}(h,U,I,S^m) = l$. We denote by $S^l|_{(h,U,I)}$ the sub-sample with $[x \in U, h(x) \in I]$, and its size is $l$.

Now, in order to use Chernoff inequality, we define two random variables, $\hat{Z}_y(h,U,I)$ and $\hat{Z}_h(h,U,I)$, as follows:

$$\hat{Z}_y(h,U,I) := \frac{1}{l} \sum_{(x_i,y_i) \in S^l|_{(h,U,I)}} y_i$$

$$\hat{Z}_h(h,U,I) := \frac{1}{l} \sum_{(x_i,y_i) \in S^l|_{(h,U,I)}} h(x_i)$$

and we observe that

$$\mathbb{E}\left[ \hat{Z}_y(h,U,I) \right] = \mu_h(h,U,I)$$

$$\mathbb{E}\left[ \hat{Z}_h(h,U,I) \right] = \mu_y(h,U,I)$$

Using this notation,

$$\Pr\left[ S^m \in B^m \;\middle|\; S^m \in G^{m,l} \right]$$

$$\leq \sum_{h \in \mathcal{H}} \sum_{(U,I) \in C_h} \left[ \Pr\left[ \left| \hat{Z}_y(h,U,I) - \mu_h(h,U,I) \right| > \frac{\epsilon}{2} \right] + \Pr\left[ \left| \hat{Z}_h(h,U,I) - \mu_y(h,U,I) \right| > \frac{\epsilon}{2} \right] \right]$$

$$\leq \sum_{h \in \mathcal{H}} \sum_{(U,I) \in C_h} 4 e^{-\frac{\epsilon^2}{2} l} \leq \frac{4|\Gamma||\mathcal{H}|}{\lambda} e^{-\frac{\epsilon^2}{2} l}$$

We would like to set $l$ so that $\Pr\left[ S^m \in B^m \mid S^m \in G^{m,l} \right]$ will be at most $\delta/2$, as follows,

$$\frac{4|\Gamma||\mathcal{H}|}{\lambda} e^{-\frac{\epsilon^2}{2} l} \leq \frac{\delta}{2} \iff l \geq \frac{2}{\epsilon^2} \log\left( \frac{8|\Gamma||\mathcal{H}|}{\delta\lambda} \right)$$

Hence, we set

$$l = \frac{2}{\epsilon^2} \log\left( \frac{8|\Gamma||\mathcal{H}|}{\delta\lambda} \right)$$

Next, we will bound $\Pr\left[S^m \in \overline{G^{m,l}}\right]$ by $\delta/2$.

Since $m \geq m_{\mathcal{H}}(\epsilon, \delta, \psi, \gamma, \lambda)$ and since $p(h, U, I) \geq \gamma\psi$ for any $h \in \mathcal{H}$ and $(U, I) \in C_h$, we know that for any $h \in \mathcal{H}$ and $(U, I) \in C_h$:

$$m \geq \frac{4l}{\gamma\psi} = \frac{8\log\left(\frac{8|\Gamma||\mathcal{H}|}{\delta\lambda}\right)}{\epsilon^2\gamma\psi}$$

Thus, the expected number of samples we have in each interesting category, is at least twice the value of $l$, i.e.,

$$\mathbb{E}[\hat{n}(h, U, I, S)] = mp(h, U, I) \geq m\gamma\psi \geq 2l$$

Thus, using the relative version of Chernoff bound, the upper bound we have on $l$, and the lower bound we have on $m$, for any $h \in \mathcal{H}$ and for any interesting category $(U, I) \in C_h$, the probability that $S^m$ has less than $l$ samples in the category $(U, I)$ is bounded by:

$$\Pr[\hat{n}(h, U, I, S) \leq l] \leq \Pr\left[\hat{n}(h, U, I, S) \leq \frac{\mathbb{E}[\hat{n}(h, U, I, S)]}{2}\right] \leq e^{-\frac{\mathbb{E}[\hat{n}(h,U,I,S)]}{8}} \leq \frac{\lambda\delta}{2|\Gamma||\mathcal{H}|}$$

And, by using the union bound:

$$\Pr[S^m \in \overline{G^{m,l}}] = \Pr\left[\exists h \in \mathcal{H}, \exists (U, I) \in C_h : \hat{n}(h, U, I, S) < l\right] \leq |C_h|\frac{\lambda\delta}{2|\Gamma|} \leq \frac{\delta}{2}$$

Thus, overall:

$$\Pr[S^m \in B^m] \leq \Pr\left[S^m \in B^m \mid S^m \in G^{m,l}\right] + \Pr[S^m \in \overline{G^{m,l}}] \leq \delta/2 + \delta/2 = \delta$$

as required.

∎

## C   Proofs for Section 5

*Proof.* (Proof of Lemma 16)
Let us assume that $VCdim(\mathcal{H}_v) > d$ and let $S$ be a sample of size $d+1$ such that $\mathcal{H}_v$ shatters $S$.

Let us define the function $f : S \to \mathcal{Y}$ as:

$$\forall x \in S : f(x) = v$$

Let $T \subseteq S$ be an arbitrary subset of $S$. By assuming that $\mathcal{H}_v$ shatters $S$ we know that there exists $h_v \in \mathcal{H}_v$ such that:

$$\forall x \in S : h_v(x) = 1 \iff x \in T$$

This means that for the corresponding predictor $h \in \mathcal{H}$:

$$\forall x \in S : h(x) = v = f(x) \iff x \in T$$

Thus, using our definition of $f$,

$$\forall T \subseteq S, \exists h \in \mathcal{H}, \forall x \in S : h(x) = f(x) \iff x \in T$$

Which means that $S$ is G-shattered by $\mathcal{H}$. However, since $|S| > d$, it is a contradiction to the assumption that $d_G(\mathcal{H}) \leq d$. ∎

*Proof.* (Proof of Lemma 17)
Assume that $VCdim(\Phi_{\mathcal{H}_v}) > d$ and let $S$ be a sample of $d+1$ domain points and outcomes shattered by $\Phi_{\mathcal{H}_v}$.

Note that $y = 0$ implies that $\forall h_v \in \mathcal{H}_v, \forall x \in \mathcal{X} : \phi_{h_v}(x, y) = 0$. Thus, $\forall (x, y) \in S : y = 1$ (otherwise $S$ cannot be shattered).

Let $S_x = \{x_j : (x_j, y_j) \in S\}$. Observe that when $y = 1$, $\forall h_v \in \mathcal{H}_v, \forall x \in \mathcal{X} : \phi_{h_v}(x, 1) = h_v(x)$. Thus, the fact that $S$ is shattered by $\Phi_{\mathcal{H}_v}$ implies that $S_x$ is shattered by $\mathcal{H}_v$. However, $|S_x| = d + 1$. Thus, we have a contradiction to the assumption that $VCdim(\Phi_{\mathcal{H}_v}) > d$. ∎

*Proof.* (Proof of Lemma 18)

Let $\mathcal{H}_v$ and $\Phi_{\mathcal{H}_v}$ be the binary prediction and binary prediction-outcome classes of $\mathcal{H}$.

Using Lemmas 16 and 17, and since $d_G(\mathcal{H}) \leq d$, we know that $VCdim(\Phi_{\mathcal{H}_v}) \leq VCdim(\mathcal{H}_v) \leq d$.

In addition, note that:

$$\left| \frac{1}{m} \sum_{i=1}^{m} \mathbb{I}\left[h(x_i) = v\right] - \Pr_{x \sim D_U}\left[h(x) = v\right] \right| = \left| \frac{1}{m} \sum_{i=1}^{m} h_v(x_i) - \Pr_{x \sim D_U}\left[h_v(x) = 1\right] \right|,$$

And

$$\left| \frac{1}{m} \sum_{i=1}^{m} \mathbb{I}\left[h(x_i) = v, y = 1\right] - \Pr_{(x,y) \sim D_U}\left[h(x) = v, y = 1\right] \right| = \left| \frac{1}{m} \sum_{i=1}^{m} \phi_{h,v}(x_i, y_1) - \Pr_{(x,y) \sim D_U}\left[\phi_{h,v}(x, y)\right] \right|.$$

and the lemma follows directly from Corollary 13. ∎

*Proof.* (Proof of Lemma 19)

Let us denote $\xi := \psi\epsilon/3$

$$\frac{p_1}{p_2} - \frac{\tilde{p}_1}{\tilde{p}_2} \leq \frac{p_1}{p_2} - \frac{p_1 - \xi}{p_2 + \xi} = \frac{p_1(1 + \xi/p_2)}{p_2(1 + \xi/p_2)} - \frac{p_1 - \xi}{p_2(1 + \xi/p_2)} = \frac{\xi}{p_2(1 + \xi/p_2)}\left[\frac{p_1}{p_2} + 1\right]$$

Since $p_1, \psi \leq p_2$,

$$\frac{\xi}{p_2(1 + \xi/p_2)}\left[\frac{p_1}{p_2} + 1\right] \leq \frac{\xi}{p_2}\left[\frac{p_2}{\psi} + \frac{p_2}{\psi}\right] = \frac{2\xi}{\psi} \leq \frac{3\xi}{\psi} = \epsilon.$$

Similarly,

$$\frac{\tilde{p}_1}{\tilde{p}_2} - \frac{p_1}{p_2} \leq \frac{p_1 + \xi}{p_2 - \xi} - \frac{p_1}{p_2} = \frac{p_1 + \xi}{p_2(1 - \xi/p_2)} - \frac{p_1(1 - \xi/p_2)}{p_2(1 - \xi/p_2)} = \frac{\xi}{p_2(1 - \xi/p_2)}\left[1 + \frac{p_1}{p_2}\right].$$

Since $p_1, \psi \leq p_2$,

$$\frac{\xi}{p_2(1 - \xi/p_2)}\left[1 + \frac{p_1}{p_2}\right] \leq \frac{\xi}{p_2(1 - \xi/\psi)}\left[\frac{p_2}{\psi} + \frac{p_2}{\psi}\right] = \frac{2\xi}{\psi(1 - \xi/\psi)} = \frac{2\epsilon}{3(1 - \epsilon/3)} \leq \frac{2\epsilon}{3(1 - 1/3)} = \epsilon$$

Thus,

$$\left| \frac{p_1}{p_2} - \frac{\tilde{p}_1}{\tilde{p}_2} \right| \leq \epsilon$$

∎

*Proof.* (Proof of Lemma 20) Let $\mathrm{P}_U$ denote the probability of subpopulation $U$:

$$\mathrm{P}_U := \Pr_{x \sim D}[x \in U]$$

Using the relative Chernoff bound (Lemma 23) and since $\mathbb{E}[|S \cap U|] = m\mathrm{P}_U$, we can bound the probability of having a small sample size in $U$. Namely, if $\mathrm{P}_U \geq \gamma$, then:

$$\Pr_D\left[|S \cap U| \leq \frac{\gamma m}{2}\right] \leq \Pr_D\left[|S \cap U| \leq \frac{m\mathrm{P}_U}{2}\right] \leq e^{-\frac{m\mathrm{P}_U}{8}} \leq e^{-\frac{\gamma m}{8}}$$

Thus, for any $U \in \Gamma_\gamma$, if $m \geq \frac{8\log\left(\frac{|\Gamma|}{\delta}\right)}{\gamma}$, then, with probability of at least $1 - \frac{\delta}{|\Gamma|}$,

$$|S \cap U| > \frac{\gamma m}{2}$$

Finally, using the union bound, with probability at least $1 - \delta$, for all $U \in \Gamma_\gamma$,

$$|S \cap U| > \frac{\gamma m}{2}$$

∎

*Proof.* (Proof of Theorem 10)

Let $S = \{(x_1, y_1), ..., (x_m, y_m)\}$ be a sample of $m$ labeled examples drawn i.i.d. according to $D$, and let $S_U := \{(x, y) \in S : x \in U\}$ be the samples in $S$ that belong to subpopulation $U$.

Let $\Gamma_\gamma$ denote the set of all subpopulations $U \in \Gamma$ that has probability of at least $\gamma$:

$$\Gamma_\gamma := \{U \in \Gamma \mid \Pr_{x \sim D}[x \in U] \geq \gamma\}$$

Let us assume the following lower bound on the sample size:

$$m \geq \frac{8 \log\left(\frac{2|\Gamma|}{\delta}\right)}{\gamma}$$

Thus, using Lemma 20, we can bound the probability of having a subpopulation $U \in \Gamma_\gamma$ with small number of samples. Namely, we know that with probability of at least $1 - \delta/2$, for every $U \in \Gamma_\gamma$:

$$|S_U| \geq \frac{\gamma m}{2}$$

Next, we would like to show that having a large sample size in $U$ implies accurate approximation of the calibration error, with high probability, for any interesting category in $(U, I)$. For this purpose, let us define $\epsilon', \delta'$ as:

$$\epsilon' := \frac{\psi \epsilon}{3}$$

$$\delta' := \frac{\delta}{4|\Gamma||\mathcal{Y}|}$$

By using Lemma 18 and since $d_G(\mathcal{H}) \leq d$, we know that there exists some constant $a > 0$, such that, for any $v \in \mathcal{Y}$ and any $U \in \Gamma_\gamma$, with probability at least $1 - \delta'$, a random sample of $m_1$ examples from $U$, where,

$$m_1 \geq a \frac{d + \log(1/\delta')}{\epsilon'^2} = 9a \frac{d + \log(\frac{4|\Gamma||\mathcal{Y}|}{\delta})}{\epsilon^2 \psi^2}$$

will have,

$$\forall h \in \mathcal{H} : \left| \frac{1}{m_1} \sum_{x' \in S_U} \mathbb{I}\left[h(x') = v\right] - \Pr\left[h(x) = v \mid x \in U\right] \right| \leq \epsilon' = \frac{\psi \epsilon}{3}$$

By using Lemma 18 and since $d_G(\mathcal{H}) \leq d$, we know that for any $v \in \mathcal{Y}$ and any $U \in \Gamma_\gamma$, with probability at least $1 - \delta'$, a random sample of $m_2$ labeled examples from $U \times \{0, 1\}$, where,

$$m_2 \geq a \frac{d + \log(1/\delta')}{\epsilon'^2} = 9a \frac{d + \log(\frac{4|\Gamma||\mathcal{Y}|}{\delta})}{\epsilon^2 \psi^2}$$

will have,

$$\forall h \in \mathcal{H} : \left| \frac{1}{m_2} \sum_{(x', y') \in S_U} \mathbb{I}\left[h(x') = v, y' = 1\right] - \Pr\left[h(x) = v, y = 1 \mid x \in U\right] \right| \leq \epsilon' = \frac{\psi \epsilon}{3}$$

Let us define the constant $a'$ in a manner that sets an upper bound on both $m_1$ and $m_2$:

$$a' := 18a$$

and let $m'$ be that upper bound:

$$m' := a' \frac{d + \log\left(\frac{|\Gamma||\mathcal{Y}|}{\delta}\right)}{\psi^2 \epsilon^2} \geq \max(m_1, m_2)$$

Then, by the union bound, if for all subpopulation $U \in \Gamma_\gamma$, $|S_U| \geq m'$, then, with probability at least $1 - 2|\Gamma||\mathcal{Y}|\delta' = 1 - \frac{\delta}{2}$:

$$\forall h \in \mathcal{H}, \forall U \in \Gamma_\gamma, \forall v \in \mathcal{Y}:$$

$$\left| \frac{1}{|S_U|} \sum_{(x',y') \in S_U} \mathbb{I}\left[h(x') = v\right] - \Pr\left[h(x) = v \mid x \in U\right] \right| \leq \frac{\psi\epsilon}{3}$$

$$\forall h \in \mathcal{H}, \forall U \in \Gamma_\gamma, \forall v \in \mathcal{Y}:$$

$$\left| \frac{1}{|S_U|} \sum_{(x',y') \in S_U} \mathbb{I}\left[h(x') = v, y' = 1\right] - \Pr\left[h(x) = v, y = 1 \mid x \in U\right] \right| \leq \frac{\psi\epsilon}{3}$$

Let us choose the sample size $m$ as follows:

$$m := \frac{2m'}{\gamma} = 2a \frac{d + \log\left(\frac{|\Gamma||\mathcal{Y}|}{\delta}\right)}{\psi^2 \epsilon^2 \gamma}$$

Recall that with probability at least $1 - \delta/2$, for every $U \in \Gamma_\gamma$:

$$|S_U| \geq \frac{\gamma m}{2} = m'$$

Thus, using the union bound once again, with probability at least $1 - \delta$:

$$\forall h \in \mathcal{H}, \forall U \in \Gamma_\gamma, \forall v \in \mathcal{Y}:$$

$$\left| \frac{1}{|S_U|} \sum_{x' \in S_U} \mathbb{I}\left[h(x') = v\right] - \Pr\left[h(x) = v \mid x \in U\right] \right| \leq \frac{\psi\epsilon}{3}$$

$$\forall h \in \mathcal{H}, \forall U \in \Gamma_\gamma, \forall v \in \mathcal{Y}:$$

$$\left| \frac{1}{|S_U|} \sum_{(x',y') \in S_U} \mathbb{I}\left[h(x') = v, y' = 1\right] - \Pr\left[h(x) = v, y = 1 \mid x \in U\right] \right| \leq \frac{\psi\epsilon}{3}$$

To conclude the theorem, we need show that having $\psi\epsilon/3$ approximation to the terms described above, implies accurate approximation to the calibration error. For this purpose, let us denote:

$$p_1(h, U, v) := \Pr\left[h(x) = v, y = 1 \mid x \in U\right]$$
$$p_2(h, U, v) := \Pr\left[h(x) = v \mid x \in U\right]$$
$$\tilde{p}_1(h, U, v) := \frac{1}{|S_U|} \sum_{(x',y') \in S_U} \mathbb{I}\left[h(x') = v, y' = 1\right]$$
$$\tilde{p}_2(h, U, v) := \frac{1}{|S_U|} \sum_{x' \in S_U} \mathbb{I}\left[h(x') = v\right]$$

Then, with probability at least $1 - \delta$:

$$\forall h \in \mathcal{H}, \forall U \in \Gamma_\gamma, \forall v \in \mathcal{Y}: \left| \tilde{p}_2(h, U, v) - p_2(h, U, v) \right| \leq \frac{\psi\epsilon}{3}$$

$$\forall h \in \mathcal{H}, \forall U \in \Gamma_\gamma, \forall v \in \mathcal{Y}: \left| \tilde{p}_1(h, U, v) - p_1(h, U, v) \right| \leq \frac{\psi\epsilon}{3}$$

Using Lemma 19, for all $h \in \mathcal{H}$, $U \in \Gamma_\gamma$ and $v \in \mathcal{Y}$, if $p_2(h, U, v) \geq \psi$, then:

$$\left| \frac{p_1(h, U, v)}{p_2(h, U, v)} - \frac{\tilde{p}_1(h, U, v)}{\tilde{p}_2(h, U, v)} \right| \leq \epsilon$$

Thus, since

$$c(h, U, \{v\}) = \frac{p_1(h, U, v)}{p_2(h, U, v)} - v$$

$$\hat{c}(h, U, \{v\}, S) = \frac{\tilde{p}_1(h, U, v)}{\tilde{p}_2(h, U, v)} - v$$

then with probability at least $1 - \delta$:

$$\forall h \in \mathcal{H}, \forall U \in \Gamma, \forall v \in \mathcal{Y}: \quad \Pr[x \in U] \geq \gamma, \Pr[h(x) = v \mid x \in U] \geq \psi \Rightarrow |c(h, U, \{v\}) - \hat{c}(h, U, \{v\}, S)| \leq \epsilon$$

■

## D    Proofs for Section 6

*Proof.* (Proof of Theorem 11) Let $\mathcal{X} = U \cup \{x^2\}$ where $U = \{x^0, x^1\}$ and $x^0 \neq x^1$. Let $H = \{h\}$, where

$$h(x) = \begin{cases} \frac{1}{2} + \epsilon & x = x^0 \\ 0 & else. \end{cases}$$

Let $\Gamma = \{U, \{x^2\}\}$. Let $D \in \{D_1, D_2\}$ where

$$D_1(x, y) = \begin{cases} (1/2 + \epsilon)\psi\gamma & (x, y) = (x^0, 1) \\ (1/2 - \epsilon)\psi\gamma & (x, y) = (x^0, 0) \\ (1 - \psi)\gamma & (x, y) = (x^1, 0) \\ 1 - \gamma & (x, y) = (x^2, 0) \end{cases}$$

and

$$D_2(x, y) = \begin{cases} (1/2 + \epsilon)\psi\gamma & (x, y) = (x^0, 0) \\ (1/2 - \epsilon)\psi\gamma & (x, y) = (x^0, 1) \\ (1 - \psi)\gamma & (x, y) = (x^1, 0) \\ 1 - \gamma & (x, y) = (x^2, 0) \end{cases}$$

Now we will show a reduction to coin tossing:
Consider two biased coins. The first coin has a probability of $r_1 = 1/2 + \epsilon$ for heads and the second has a probability of $r_2 = 1/2 - \epsilon$ for heads. We know that in order to distinguish between the two with confidence $\geq 1 - \delta_1$, we need at least $C \frac{\ln(\frac{1}{\delta_1})}{\epsilon^2}$ samples.

Since

$$\Pr_{(x,y) \sim D}[x \in U] = \Pr_{(x,y) \sim D}[x \neq x^2] = \gamma$$

the first condition for multicalibration holds. Now, we use another property of our "tailor-maded" distribution $D$ and single predictor $h$, which is $\{x \in \mathcal{X} : h(x) = \frac{1}{2} + \epsilon\} = \{x \in \mathcal{X} : h(x) = \frac{1}{2} + \epsilon, x \in U\} = \{x_0\}$, to get the second condition:

$$\Pr_D[h(x) = 1/2 + \epsilon | x \in U] = \Pr_D[x = x^0 | x \in U] = \frac{\psi\gamma}{\gamma} = \psi,$$

and that

$$\Pr_D[y = 1 | h(x) = \frac{1}{2} + \epsilon, x \in U] = \Pr_D[y = 1 | x = x^0]$$

is either $1/2 + \epsilon$ (if $D = D_1$) or $1/2 - \epsilon$ (in case $D = D_2$) (recall that $D \in \{D_1, D_2\}$).

Now, if $H$ has the multicalibration uniform convergence property with a sample $S = (x_i, y_i)_{i=1}^m$ of size $m$, and if

$$\sum_{i=1}^m \frac{\mathbb{I}[y_i = 1, h(x_i) = 1/2 + \epsilon, x_i \in U]}{\sum_{j=1}^m \mathbb{I}[h(x_i) = 1/2 + \epsilon, x_i \in U]} = \sum_{i=1}^m \frac{\mathbb{I}[y_i = 1, x_i = x^0]}{\sum_{j=1}^m \mathbb{I}[x_i = x^0]} > \frac{1}{2}$$

holds, then

$$\Pr[y = 1 | h(x) = \frac{1}{2} + \epsilon, x \in U] = \frac{1}{2} + \epsilon$$

holds w.p. $1 - \delta_1$ (from the definition of multicalibration uniform convergence).

Let us assume by contradiction that we can get multicalibration uniform convergence with $m = \frac{C}{\epsilon^2 \psi \gamma} - \frac{k}{\psi \gamma} < \frac{C}{\epsilon^2 \psi \gamma}$ for some constant $k = \Omega(1)$.

Let $m_0$ denote the random variable that represents the number of samples in $S$ such that $x_i = x^0$ (i.e., $h(x_i) = 1/2 + \epsilon$). Hence, $\mathbb{E}[m^0] = \gamma \cdot \psi \cdot m = \frac{C}{\epsilon^2} - k$.

From Hoeffding's inequality,

$$\Pr[m^0 \geq \frac{C}{\epsilon^2}] = \Pr[m^0 - \underbrace{(\frac{C}{\epsilon^2} - k)}_{\mathbb{E}[m_0]} \geq k] \leq e^{-2mk^2}.$$

Let $\delta_2$ be the parameter that holds $e^{-2mk^2} \leq \delta_2$, and let $\delta := \delta_1 + \delta_2$. Then we get that with probability $> (1 - \delta_1)(1 - \delta_2) > 1 - \delta_1 - \delta_2 = 1 - \delta$ we can distinguish between the two coins with less than $\frac{C}{\epsilon^2}$ samples, which is a contradiction.

■