[Reviews · NeurIPS 2020]

Review 1

Summary and Contributions: This paper provides convergence bounds on multicalibration error, for both finite and infinite predictor classes.

Strengths: This paper is very thorough and fairly easy to follow, even for a non-expert. The authors provide sufficient context for their work; their notation is clear and consistent; most important definitions are contained in the main paper; the main results are clearly presented and discussed briefly; while all proofs are in the appendix, the proof sketches are cliear and helpful.

Weaknesses: There is little discussion of these results and their potential impact on the field -- and specifically, on "fairness" in machine learning. It appears that this work is motivated by fairness or justice, though the authors only briefly discuss these issues.

Correctness: The results are intuitive, and the proof sketches are clear and seem correct. I did not verify any of the proofs in the appendices.

Clarity: This paper is easy to follow, and uses consistent notation. (Much appreciated!)

Relation to Prior Work: Yes

Reproducibility: Yes

Additional Feedback:


Review 2

Summary and Contributions: This paper studies the problem of understanding the sample complexity required for uniform convergence guarantees of the multicalibration error over a function class. The sample complexity bounds in this paper focus on groups with "large" sizes. Update Post-rebuttal: I am satisfied with the authors' response to my queries during the rebuttal phase and would like to keep my rating the same.

Strengths: The problem of understanding sample complexity of multicalibration error is quite relevant to the ML community, with the growing emphasis on the understanding the group fairness properties of classifiers. This main contributions of the paper is to provide such sample complexity bounds in terms of the graph dimension of the underlying function class. In addition, the paper also provides a worst-case lower bound exhibiting dependence on relevant quantities.

Weaknesses: I would like to highlight a few questions/remarks about the results: 1. Comparison with [1]: Are the upper bounds of [1] also valid only for the class of "interesting categories" with large enough mass or are they under a different assumption? 2. Why does the dependence on \psi change from \psi^{-1} to \psi^{-2} going from a finite class to infinite function classes? 3. Instance optimality: Are the VC dimension bounds on the predictor class and the true positive class (Lemmas 16 and 17) tight for each instance? Or can there be problem instances where these VC dimensions can be much smaller and one can expect lower sample complexity. It would be nice to see such examples if they exist. 4. Can the authors comment on the applicability of the proposed techniques for studying generalization bounds for complex measures like F-score which appear to have similar challenges since they also comprise ratio of empirically estimated quantities? 5. In addition to only the multicalibration error, what are the challenges in extending such an analysis to constrained or regularized objectives which contain both a loss component as well as a calibration error component? 6. Dimension dependence in lower bound - Is it possible to get lower bounds for every d>0 and show that the graph dimension factor is indeed necessary? [1] Hebert-Johnson, U., Kim, M., Reingold, O., & Rothblum, G. (2018). Multicalibration: Calibration for the (Computationally-identifiable) masses.

Correctness: The theoretical claims in the paper appear mostly correct under the stated assumptions.

Clarity: Overall, the paper is quite well written and easy to follow. However, I found that the statement of the lower bound (theorem 11) is imprecise in the way it is currently stated. It does not mention if the bound holds for all function classes or is it for some function class; it would be helpful if more details are added to quantify the dependence on the parameters.

Relation to Prior Work: The paper contextualize their results in terms of existing works and provide a detailed discussion of how it compres with respect to these.

Reproducibility: Yes

Additional Feedback:


Review 3

Summary and Contributions: This article deals with group fairness. The authors introduce three bounds (sample complexities) for the multicalibration error. Two upper bounds, namely Theorems 9 and 10, are respectively dedicated to function classes of finite cardinality and function classes of finite graph dimension. The third uniform convergence result is a lower bound: Theorem 11.

Strengths: The contribution appears technically sound. The bounds are globally sharper than the sample bound of reference, obtained by Hebert-Johnson et al. (2018). They are also more general.The authors found original solutions to handle the specificities of multicalibration learning (compared to the standard VC theory).

Weaknesses: The impact of the contribution could be limited given the hypotheses made. What about studying the case of multi-category pattern classification, or margin classifiers?

Correctness: The claims seem correct (I could check all proofs).

Clarity: The paper has significantly improved since it was submitted to ICML 2020. However, some typos have not been corrected. One example, on line 345 : VC-diemnsion -> VC-dimension. Furthermore, there are new ones, for example on line 164, where two spaces are missing.

Relation to Prior Work: The comparison with the contribution of Hebert-Johnson et al. (2018) is satisfactory.

Reproducibility: Yes

Additional Feedback:


Review 4

Summary and Contributions: POST REBUTTAL UPDATE During the initial review I had been vacillating between a score of 6 and 7. After reading the other reviews and the rebuttal, I am bumping this up to 7 since the paper is certainly interesting, and I would rather err on the side of its inclusion. ----------- This paper is concerned with the sample complexity of multicaliberation uniform convergence, a notion that has gained renewed relevance due to applications in fair ML. The notion of caliberation for a subpopulation and an output range demands that the classifier is roughly faithful to the true average label given the catergory and its output range. The multi- corresponds to having to be caliberated on many subpopulation-output interval pairs (called categories). The problem is challenging because the notion of caliberation error requires conditioning on the output values of a hypothesis, which precludes naive use of standard bounds. The authors first give a natural definition of multicaliberation uniform convergence - the principle ingredient is a notion of `interesting' category - those such that the subpopulation, and the output interval given the subpopulation, both have nontrivial mass. The main results are two upper bounds on the sample complexity of mult.cal. unif. conv. the first (Thm 9) is for the setting of finite hypothesis classes, and catergories where intervals correspond to [j\lambda, j\lambda+1). The second (Thm. 10) is for the setting of hypothesis classes which have finite graph dimension, and have function outputs in a finite set. These are complemented by a (restricted) lower bound (Thm 11) that shows tightness in some parameters.

Strengths: The main strengths, and indeed main contributions of the paper are the two upper bounds on sample complexity of multicaliberation unfirom convergence. While I find Thm. 9 quite natural and simple, it gains importance in that it improves upon upper bounds relative to the existing literature, in that it has tighter behaviour on the error (\epsilon) and the interestingness (\psi, \gamma) parameters. Thm. 10, on the other hand, I find quite interesting. The use of the graph dimension is clever, and the idea is well executed. On the whole, I find the paper interesting. I think it nicely details and represents progress on a relevant problem, and has nontrivial upper bounds in Thm 10.

Weaknesses: The glaring weakness of the paper is that the argument for tightness of the upper bounds is very weak, and that no characterisation of what the correct notion of complexity of a class w.r.t. multi.cal. unif. conv. is determined. The only lower bound in the paper is that of Thm 11. To begin, I think the statement of this theorem is too strong, and its claims should be written more carefully. The proof presented only shows that there exists a single function (with carefully picked output values) and one (small) set of catergories for which Omega(.) samples are required. To claim that "for multicaliberation unif. conv. ... we need \Omega(.) samples" sounds way too strong. More importantly, the bound doesn't capture any features of the complexity of the catergories (number of subpopulations and discretisation level). Furthermore, it doesn't capture any notion of the complexity of the hypothesis class H, and thus doesn't hold much weight as a lower bound in my opinion. Indeed, even though I find the paper interesting, the above limits how strongly I can recommend it, and thus is the main reason for the overall rating below.

Correctness: Yes. Although I would like the statement of Thm. 11 to be altered.

Clarity: The paper is well written. The problem is motivated well, and contextualised nicely, and care is taken to describe the technical challenge of the problem. Additionally, the proof sketches and development in the main text do a good job of explaining the key ideas.

Relation to Prior Work: Yes.

Reproducibility: Yes

Additional Feedback: -- Line 364 (Finite v/s Cont): Doesn't the \alpha in (\alpha, \psi, \gamma)-multicaliberation give a natural scale of discretisation of the problem? In fact, considering this suggests that a O(\alpha)-fat shattering dimension may be a relevant complexity measure. -- This is maybe obvious, but I think it's worth stating that multical. unif. conv. is not formally necessary for learning multicaliberated predictors. -- Line 41 and 44 make contradictory statements - the first says that one may want to trade off prediction and caliberation errors, while the second says that there's no reason to prefer a uncaliberated predictor to a caliberated one. Please smooth this out.

[Author Response · NeurIPS 2020]

We thank the reviewers for their thorough and positive reviews. Overall, we were glad to see the reviewers found the paper to be clearly written, technically sound, and the results to be of interest to the (fair) ML community.

We will of course incorporate all the suggested edits by the reviewers as well as more clarifications.

**General comments:**

Regarding our lower bound: Our lower bound holds both for finite predictor classes and infinite predictor classes with finite graph dimension. We will restate the theorem statement so it would state precisely what is proven. Indeed, it remains to be an open problem whether it can be improved, and in particular exhibit a dependency on graph dimension, which we assume is necessary.

As for other complexity measures such as fat-shattering and margin classifiers, it could be an interesting direction for future work. In this paper, we chose to derive the generalization bounds using Graph dimension and VC-dimension.

**Reviewer 1:** Thank you for your positive review. The reviewer expressed concern regarding the brevity of the discussion about the potential societal impacts of the results. We will address these important issues in detail in the final version.

**Reviewer 2:** Thanks for your careful reading and constructive questions. We address your concerns as follows:

1. The upper bounds of [Hebert-Johnson et al.] are implicitly guaranteed only for "interesting categories" with large enough mass.

2. *Dependence on $\psi$:* In the case of finite predictor classes, we applied union-bound directly on all the (large enough) "interesting categories", that have finite cardinality due to the finite number of hypotheses. In the case of infinite predictor classes, this technique is no longer possible. To derive the concentration bound of $c(h, U, v)$ we bound the deviations of $p_1(h, U, v)$ and $p_2(h, U, v)$ (see line 592, in the proof of Theorem 10 in the supplementary material). To achieve our desired accuracy, we need to bound the deviation by $\psi\epsilon/3$ (see Lemma 19, line 331). This increased accuracy translates to the $\psi^{-2}$ in the sample size.

3. *Instance optimality:* Your intuition is correct, there are some problem instances with smaller VCs for all the values of $v$ that determines the "intresting categories", thus their sample complexity is lower. On the other hand, the bound is tight in the worst case. Namely, for any graph dimension $d$, we can consider a predictor classes with $VC(\mathcal{H}_v) = VC(\Phi_{\mathcal{H}_V}) = d$ for any prediction value $v$.

4. *Applicability of the proposed techniques:* Our techniques are suitable to derive generalization bounds for complex measures such as F-score. Specifically for F-score, one can derive generalization bounds, assuming that at least one of TP, FP, or FN is "not negligible".

5. *Objectives which contain both a loss component as well as a calibration error component* Our generalization bounds extend to such settings. For example, for zero-one loss, we can guarantee simultaneously for each predictor in the class that all the empirical multicalibraion errors of "interesting categroies" and the empirical loss are close to their expectations. This does not even require an increase in the sample complexity.

**Reviewer 3:** We thank the reviewer for their positive and constructive review. As for multi-category pattern classification, it can be reduced to binary classification, which we address in this work.

**Reviewer 4:** Thank you for diving deeply into the paper and for your detailed comments and corrections. Sure, one can choose $\alpha$ to be the discretization scale, but this is a limitation we wanted to avoid in order to give more flexibility.

[Meta-Review · NeurIPS 2020]

All the reviewers found the paper interesting and well-written. Please incorporate the reviewers' suggestions in the final version.